# Transplantation of Human-Fetal-Spinal-Cord-Derived NPCs Primed with a Polyglutamate-Conjugated Rho/Rock Inhibitor in Acute Spinal Cord Injury

**DOI:** 10.3390/cells11203304

**Published:** 2022-10-20

**Authors:** Esther Giraldo, Pablo Bonilla, Mara Mellado, Pablo Garcia-Manau, Carlota Rodo, Ana Alastrue, Eric Lopez, Elena Carreras Moratonas, Ferran Pellise, Snežana Đorđević, María J. Vicent, Victoria Moreno Manzano

**Affiliations:** 1Neuronal and Tissue Regeneration Laboratory, Centro de Investigación Príncipe Felipe, E-46012 Valencia, Spain; 2Department of Biotechnology. Universitat Politècnica de València, E-46022 Valencia, Spain; 3UPV-CIPF Joint Research Unit Disease Mechanisms and Nanomedicine, Centro de Investigación Príncipe Felipe, E-46012 Valencia, Spain; 4Maternal-Foetal Medicine Unit, Vall d’Hebron Hospital Campus, E-08035 Barcelona, Spain; 5Spine Surgery Unit, Hospital Universitari Vall d’Hebron, E-08035 Barcelona, Spain; 6Polymer Therapeutics Laboratory, Centro de Investigación Príncipe Felipe, E-46012, Valencia, Spain

**Keywords:** human fetal neural precursor, NPC transplantation, Rho/ROCK kinase inhibition, cell priming, spinal cord injury

## Abstract

Neural precursor cell (NPC) transplantation represents a promising therapy for treating spinal cord injuries (SCIs); however, despite successful results obtained in preclinical models, the clinical translation of this approach remains challenging due, in part, to the lack of consensus on an optimal cell source for human neuronal cells. Depending on the cell source, additional limitations to NPC-based therapies include high tumorigenic potential, alongside poor graft survival and engraftment into host spinal tissue. We previously demonstrated that NPCs derived from rat fetal spinal cords primed with a polyglutamate (PGA)-conjugated form of the Rho/Rock inhibitor fasudil (PGA-SS-FAS) displayed enhanced neuronal differentiation and graft survival when compared to non-primed NPCs. We now conducted a similar study of human-fetal-spinal-cord-derived NPCs (hfNPCs) from legal gestational interruptions at the late gestational stage, at 19–21.6 weeks. In vitro, expanded hfNPCs retained neural features, multipotency, and self-renewal, which supported the development of a cell banking strategy. Before transplantation, we established a simple procedure to prime hfNPCs by overnight incubation with PGA-SS-FAS (at 50 μM FAS equiv.), which improved neuronal differentiation and overcame neurite-like retraction after lysophosphatidic-acid-induced Rho/Rock activation. The transplantation of primed hfNPCs into immune-deficient mice (NU(*NCr)-*Foxn1^nu^) immediately after the eighth thoracic segment compression prompted enhanced migration of grafted cells from the dorsal to the ventral spinal cord, increased preservation of GABAergic inhibitory Lbx1-expressing and glutamatergic excitatory Tlx3-expressing somatosensory interneurons, and elevated the numbers of preserved, c-Fos-expressing, activated neurons surrounding the injury epicenter, all in a low percentage. Overall, the priming procedure using PGA-SS-FAS could represent an alternative methodology to improve the capabilities of the hfNPC lines for a translational approach for acute SCI treatment.

## 1. Introduction

Spinal cord injury (SCI) following severe physical trauma triggers a series of complex multicellular and molecular responses, resulting in a diverse degree of permanent motor, sensory, and/or autonomic dysfunctions. Despite the promising results obtained from intense research efforts in preclinical models over the past decade, no effective therapy has been efficiently translated into the clinic. Cell transplantation and advanced cell engineering have provided hope for treatment strategies with translational potential [1,2]. Neural progenitor cell (NPC) transplantation prompts neuroprotection and induces neuroregeneration by providing neurotrophic support, attenuating secondary damage, supplying a permissive substrate for axon regrowth [3,4], replacing lost neurons and establishing novel synaptic connections with host axons [5,6], and enhancing remyelination [7,8], which ultimately promote functional recovery in rodents [9,10]. The features and origin of NPCs represent crucial aspects of developing translatable cell therapies. Primary self-renewing NPCs isolated from fetal tissue display in vitro expansion potential, thereby providing an optimal cell source, which avoids inherent variations associated with the use of different fetal donors and the cell heterogeneity, which can elevate the risk of immune rejection and/or tissue contamination. 

Standard procedures have been proposed to obtain “clinical-grade” NPCs from fetal neural tissue with minimal manipulation and in compliance with pharmaceutical good manufacturing practice (GMP) guidelines to ensure the production of advanced therapies for safe clinical use [11]. The transplantation of in vitro-expanded human fetal NPCs (hfNPCs) obtained from the lower cervical and upper thoracic spinal cord of an eight-week-old fetus prevented cyst expansion in an adult rat model of chronic SCI and post-traumatic syringomyelia [12]; however, current clinical trials with fetal tissue have employed immortalized cell lines derived from early fetal spinal cord [13] or brain tissue [14,15]. Alternative clinical approaches have employed NPCs derived from pluripotent stem cells; for example, Kumamaru et al. developed a xeno-free methodology to culture human spinal neural stem cells (NSCs) differentiated from embryonic stem cells (ESCs), thereby providing a scalable source for clinical translation [16]. These NSCs integrated into host tissue, induced the regeneration of the injured corticospinal tract, and enabled the extension of prolonged and persistent axonal projections, resulting in improved motor function in nude rats [16]. In non-human primates, these NSCs survived for nine months after transplantation and extended hundreds of thousands of human axons through monkey white matter, which established synapses and improved forelimb function, overcoming the immune graft rejection of the exogenous human cells [9]. Sugai et al. recently reported on the first clinical trial using NPCs derived from human induced pluripotent stem cells (iPSCs) as an SCI treatment [17]. Of note, while immortalized cell lines and pluripotent-cell-derived NPCs provide an inexhaustible source of cells, their application entails a high risk of tumorigenesis [18]. 

Here, we propose in vitro-amplified hfNPCs derived from the spinal cords of electively aborted fetuses at 19–21.6 weeks of gestation as an innovative approach for SCI treatment. As we recently demonstrated that priming fetal rat NPCs with PGA-SS-FAS, a polyglutamic-acid (PGA)-conjugated form of the Rho/Rock inhibitor fasudil (FAS), which provides improved stability and controlled release, enhanced graft survival and improved cell migration through the injured spinal cord [19], we also evaluated the impact of PGA-SS-FAS priming on hfNPCs. Transplantation of PGA-SS-FAS-primed hfNPCs in the acute stage of compressive SCI promoted host neuron preservation and the increased expression of c-fos, a hallmark for cell activation, without any sign of tumorigenesis.

## 2. Materials and Methods

### 2.1. Isolation and Expansion of hfNPCs

hfNPCs were isolated and expanded from human fetal spinal cord tissue obtained from five fetuses from legally elective abortions induced by vaginal Misoprostol administration after being diagnosed with severe congenital malformations at 19.0–21.6 gestational weeks of gestation at the maternal–fetal medicine department at Vall d’Hebron Hospital Campus (Barcelona, Spain). The experimental procedures were evaluated and accepted by the clinical ethical committee at the Vall d’Hebron Hospital with the approved protocol PR(AMI)120/2017. Written informed consent for anonymized tissue collection was signed by each donor. Samples with an identified central-nervous-system-associated anomaly during the ultrasound examination were excluded. Negative serology for hepatitis and AIDS from donors was confirmed. 

Human fetal spinal cords were dissected with sterile gloves and dissecting tools during the first hour after delivery. Each spinal cord, from cervical to lumbar segments, was transferred and maintained in 4 °C Hibernate™-E CTS™ Medium (Gibco™, Waltham, MA, USA) supplemented with 2% CTS™ B-27™ Supplement (Gibco™) in a hermetic sterile tube on ice for up to 5 h. In biosafe flow hoods, the spinal cords were washed twice with fresh hibernation media, and the meninges, dura, and pia mater were carefully removed, cut into ~1 mm^3^ pieces, and mechanically dissociated by repeated pipetting. The obtained cell suspension was centrifuge, and the cell pellet plated on ultra-low attachment plates in proliferation culture media—CTS™ Neurobasal™ Medium (Gibco™) supplemented with CTS™ GlutaMAX™-I Supplement (Gibco™), CTS™ B-27™ XenoFree Supplement, 100 μg/mL penicillin–streptomycin (Sigma-Aldrich; Darmstadt, Germany), 0.7 U/mL heparin (Sigma-Aldrich), 20 ng/mL human recombinant epidermal growth factor (hEGF; Peprotech, London, U.K.), 20 ng/mL basic human recombinant fibroblast growth factor (hbFGF; Peprotech), and 10 ng/mL human recombinant leukemia inhibitory factor (hLIF; Peprotech). Plates were incubated in 5% CO_2_ at 37 °C for two to three days until neurospheres formed. hfNPCs effectively formed neurospheres, which supported selection, clonal division, and cell proliferation. Neurospheres were enzymatically and mechanically dissociated using Accutase (STEMCELL™, Vancouver, Canada), and individualized cells were seeded at a density of 2.6 × 10^4^ cells/cm^2^ in six-well culture plates coated with human recombinant Laminin 521 (BioLamina, Sundbyberg, Sweden). Subsequential sub-cultivation and expansion were performed for up to an additional nine passages. hfNPC cryopreservation as neurospheres or adherent growing cultures was performed using CryoStor^®^ (Merck, Darmstadt, Germany). hfNPCs were then stored in liquid nitrogen.

### 2.2. Priming hfNPCs with PGA-SS-FAS Prior to Transplantation

Passage 5–7 hfNPCs were thawed and cultured in neurosphere-forming conditions in ultra-low attachment plates for 24 h and then incubated with PGA-SS-FAS (50 µM FAS-equiv.) for an additional 24 h in 5% CO_2_ and 37 °C. PGA-SS-FAS was synthesized and fully characterized as previously reported [19] following ICH guidelines to achieve an endotoxin-free nanoconjugate with a reliable impurity profile. The guidelines for good manufacturing practices (GMP) manufacturers were provided by PTS S.L. (Valencia, Spain). In general, PGA-SS-FAS was highly pure, with no significant impurities of residual solvents or ionic impurities. Before transplantation, hfNPCs were harvested and centrifuged at 200× *g* for 5 min and washed twice with culture media. The cell media discarded at every wash step was collected for inspection of the extracellular content of fasudil by liquid chromatography-tandem mass spectrometry (LC-MS/MS). The sample preparation and LC-MS/MS method for fasudil quantification were previously developed and described in Giraldo et al. [19]. Nevertheless, the developed LC-MS/MS method and extraction protocol were re-evaluated for linearity, the limit of quantification (LOQ), the limit of detection (LOD), recovery, and matrix effects to ensure its status as fit for purpose. Linearity was evaluated by constructing a calibration curve obtained by the internal standard method (ranitidine as an internal standard). LOD and LOQ values were calculated from the LINEST function. At the same time, matrix effects were evaluated by analyzing two different sample types: (i) blank cell medium spiked after sample preparation with fasudil as three quality control (QC) samples (low, medium, and upper) and with 1 ng/mL of ranitidine and (ii) water solution of fasudil and ranitidine in the same concentration as spiked samples. Recovery was assessed at the three QC levels by comparing the mass of extracted analyte and analyte in the sample (plasma spiked after the sample preparation) and represented as a percentage. Fasudil was not encountered in any cell washing media, suggesting that hfNPCs did not secrete fasudil or PGA-SS-FAS during washing and confirming that the cell preparation will not deliver fasudil or PGA-SS-FAS to the extracellular space or surrounding cells after transplantation into the spinal cord tissue.

### 2.3. hfNPC Proliferation Assay

The doubling time, in hours, of hfNPCs was studied by subculturing passage 1 cells in adherent conditions at a density of 1.5 × 10^4^ cells/cm^2^ until confluency, according to the following formula, *t* = t × log2/(logN_t_ − logN_0_), where t is the culture time in hours, N_0_ is the initial cell number, and N_t_ is the harvested cell number.

The number of individual proliferating cells in the cell population was assayed by adding 10 µM 5′-Br-2′-deoxyuridine (BrdU, Sigma-Aldrich) 24 h before analysis. Cells were fixed with 2% paraformaldehyde (PFA) for 10 min, washed with phosphate buffer solution (PBS), treated with 2M HCl for 20 min at room temperature, and then stopped with 0.1 M Na_2_B_4_O_7_ for 30 min. The expression of Ki67 was assayed by immunocytochemistry in the same cultures to evaluate the number of cells undergoing the G1 to mitosis transition. Cells were blocked for 1 h in Tris-buffered saline (TBS) containing 0.2% Triton-X-100 and 5% normal goat serum and then incubated with anti-BrdU (1:400; Sigma-Aldrich) and anti-Ki67 (1:400; GTX16667, GeneTex) antibodies diluted in blocking solution overnight at 4 °C. Cells were subsequently incubated with Alexa-Fluor-conjugated antibodies for 2 h at room temperature, washed twice with PBS, and finally, incubated with DAPI for nuclear staining.

### 2.4. Spontaneous hfNPC Differentiation Assay 

Neurosphere hfNPC cultures at passage 1 were dissociated with Accutase and seeded onto human recombinant Laminin 521-coated (BioLamina, Sundbyberg, Sweden) coverslips in DMEM/F12 supplemented with 100 U/mL penicillin, 100 μg/mL streptomycin, 2 mM l-glutamine, 5 mM HEPES buffer, 0.125% NaHCO_3_, 0.6% glucose, 0.025 mg/mL insulin, 80 μg/mL apotransferrin, 16 nM progesterone, 60 μM putrescine, 24 nM sodium selenite, and 2% of human serum for seven days. hfNPCs were then treated with 50 µM of PGA-SS-FAS on Day 3 to determine any effect of PGA-SS-FAS on cell differentiation. Cells were fixed on Day 7, and the neuronal, astrocytic, and oligodendroglia identity of the cells were evaluated by immunostaining with beta-III tubulin, GFAP, and OLIG2 antibodies, respectively.

### 2.5. Neurite Elongation Assays

hfNPCs in growth media were pre-treated with or without 50 µM of PGA-SS-FAS for 24 h. 10 µM lysophosphatidic acid, a pharmacological activator of the Rho/ROCK pathway (LPA; Sigma-Aldrich), was added for an additional 24 h to induce neurite retraction. Cultures were then fixed and immunolabeled with Nestin, and neurite outgrowth was quantified using the NeuronJ plug-in from ImageJ v1.48 [20].

### 2.6. Immunostaining

Cells were fixed with 4% PFA in PBS for 10 min, then permeabilized and blocked with 5% normal goat serum (NGS; Thermo Fisher) and 0.2% Triton X-100 (Sigma) and incubated overnight at 4 °C with primary antibodies. Fixed spinal cord tissues were first cryoprotected in 30% sucrose overnight at 4 °C before inclusion in Tissue-Teck OCT (Sakura Finetek Europe BV, Flemingweg, Netherlands) and then cryo-sectioned to provide 20 µm-thick sections. Tissue sections were permeabilized and blocked in PBS containing 0.1% Triton X-100, 5% horse serum, and 10% fetal bovine serum for 1 h at room temperature and then incubated overnight at 4 °C with primary antibodies.

The employed primary antibodies were: chicken anti-GFAP (1:1000; PA1-10004,Thermo Fisher), guinea pig anti-DCX (1:400; ab5910, Chemicon, Temecula, California, United States), anti-PAX6 (1:400; PRB-278P, Biolegend, San Diego, California, USA), rabbit anti-SOX2 (1:400; MAB5326, Abcam), rabbit anti-Ki67 (1:400; GTX16667, GeneTex), mouse anti-Nestin (1:400; MAB5326, Sigma Aldrich), mouse anti-Notch-1 (1:400; AF1057, R&D System, Minneapolis, Minnesota, USA), mouse anti-Olig2 (1:400; AB9610, Sigma-Aldrich), mouse anti-FOXJ1 (1:300; 14-9965-82, Thermo Fisher), mouse anti-Neurogenin1 (1:400; sc-100332, Santa Cruz), chicken anti-NeuN (1:400; ABN91, Sigma-Aldrich), rabbit anti-Tlx3 (1:5.000 gift of C. Birchmeier), guinea pig anti-Lbx1 (1:10,000 gift of C. Birchmeier), rabbit anti-c-Fos (1:400; ab190289, Abcam), and chicken anti-GFP (1:400; ab13970, Abcam).

After three washes with PBS, cells or tissues were incubated with AlexaFluor-488, -555, or -647 (1:400, Invitrogen) secondary antibodies for 2 h at room temperature. Nuclei staining was performed by incubation with DAPI (1:1000, Sigma).

Fluorescent images were acquired using an apotome fluorescent microscope (Zeiss) or confocal microscope SP8 (Leica) as indicated. Consistent exposures were applied, and images were visualized and quantified with ImageJ/Fiji software v1.48. 

Grafted cells found along the spinal cord were enumerated to analyze hfNPC survival, with the figure normalized to the microns of tissue thickness analyzed. The quantification of activated hfNPCs (double-positive for GFP and c-Fos) and the percentage of hfNPCs positive for c-Fos used a similar procedure. The neuroprotective properties of cell grafts were determined by analyzing neuronal preservation and activation 2 mm rostral and caudal to the lesion site and the lesion site (corresponding to the transplanted and injured area). Neural preservation was measured by enumerating the number of NeuN^+^ neurons, while neuronal activation used the number of neurons double-positive for NeuN and c-Fos normalized to the total analyzed area. All stainings and cell quantifications were performed from every fifth horizontal section of 20 mm in thickness in 3 animals per group. Cell number quantifications were then normalized to the analyzed summed tissue thickness (from ventral to dorsal sections) and expressed in mm^2^.

Further analysis of neuroprotective effects on specific dorsal horn somatosensory interneuron populations measured the preservation of GABAergic Ladybird homeobox 1 (Lbx1)-expressing inhibitory and glutamatergic T cell leukemia homeobox 3 (Tlx3)-expressing excitatory interneurons 2 mm rostral and 2 mm caudal to the lesion site and in the lesion site normalized to the total analyzed area. 

### 2.7. Spinal Cord Injury, hfNPC Transplantation, and Tissue Processing

All experimental procedures were approved by the Animal Care Committee of the Research Institute Principe Felipe (Valencia, 2020/VSC/PEA/0119, Spain) in accordance with the National Guide to the Care and Use of Experimental Animals (Real Decreto 53/2013). 

Immune-deficient female NU (*NCr)**-***Foxn1^nu^ mice (Charles River, France) weighing ≈20 g were housed under controlled light and temperature conditions. For surgical interventions, mice received subcutaneous morphine (5 mg/kg) 30 min prior to surgery and were anesthetized with 2% isoflurane in a continuous oxygen flow of 1 L/min. Laminectomy was performed at the T8-T9 level, exposing the dorsal surface of the dura matter. Using Bonn Micro Forceps (11083-07, Fine Science Tool, Heidelberg, Germany) (0.3 mm wide), a one-second compressive spinal cord injury was performed at the T8 thoracic level, as previously described [21]. Immediately after compression, 2 µL containing 2.5 × 10^5^ hfNPCs (previously infected with pll3.1-eGFP lentivirus for eGFP ectopic expression) primed with PGA-SS-Fas (hfNPCs + PGA-SS-FAS group), vehicle (hfNPCs group), or culture medium (control group) were intramedullary injected at the epicenter of the lesion in one single point to a 1 mm depth from the dorsal side. All animals were subjected to post-surgery care consisting of manual drainage of bladders twice a day until vesical reflex was recovered and subcutaneous administration of 5 mg/kg of enrofloxacin for seven days and 0.1 mg/kg of buprenorphine twice a day for four days. One month after injury and transplantation, animals were overdosed with an intraperitoneal administration of pentobarbital and transcardially perfused with PBS, followed by 4% of PFA in 0.1 M phosphate buffer (PB, pH = 7.4). Spinal cords were collected and maintained in 0.1M PB for further cryopreservation and histological analysis. 

Hind limb motor function was evaluated for up to four weeks using the Basso Mouse Scale (BMS) locomotor rating scale [22] by videotaping the animals in an open field twice a week using a high-definition camera. Two individuals blinded to the treatment of the mice then examined and scored motor function. We did not find significant differences at any analyzed point between the three compared groups. 

In addition, 2.5 × 10^5^ hfNPCs primed with PGA-SS-Fas or vehicle were transplanted into non-injured mice at the T8 segment to evaluate tumorigenic/invasive potential one month after transplantation. hfNPCs were monitored by semiquantitative PCR for the eGFP gene expressed by transplanted hfNPCs in brain, heart, and liver samples using specific primers (fw_AAGTCGTGCTGCTTCATGTG; rv_ GACGTAAACGGCCACAAGTT). Total RNA was isolated using TRIzol^TM^ Reagent (Invitrogen), and cDNA was synthesized from 1 µg RNA using the high-capacity RNA-to-cDNA™ kit (4368814, Applied Biosystems) following the manufacturer’s instructions. For PCR reactions, 50 ng of cDNA was amplified using GoTaq^®^ DNA Polymerase (Promega, Madison, Wisconsin, EEUU), using 55 °C for primer annealing. Mouse GAPDH amplification was used as a housekeeping gene (fw_CGGTGCTGAGTATGTCGTGGAGT; rv_CGTGGTTCACACCCATCACAAA).

### 2.8. Transmission Electron Microscopy 

For electron microscopy studies, human spinal cords were fixed in 4% PFA overnight at 4 °C. After washing steps in 0.1 M PB, 200 µm coronal sections were cut on a Leica VT-1000 vibratome (Leica, Heidelberg, Germany). Sections were post-fixed with 2% osmium, rinsed, dehydrated, and embedded in Durcupan resin (Fluka, Sigma-Aldrich, St. Louis, USA). Semithin sections (1.5 µm) were cut with an Ultracut UC-6 (Leica microsystems, Wetzlar, Germany) and stained lightly with 1% toluidine blue. Finally, ultrathin sections (70–90 nm) were cut with a diamond knife, stained with lead citrate (Reynolds solution), and examined under an FEI Tecnai G2 Spirit BioTwin transmission electron microscope (ThermoFisher Scientific, Oregon, USA) using a Morada digital camera (Olympus Soft Image Solutions GmbH, Münster, Germany).

### 2.9. Statistical Analysis

Data are graphically represented as the mean ± standard error mean (SEM) and analyzed using Graph Pad Prism software. The Shapiro–Wilk normality test was performed to evaluate each dataset’s Gaussian distribution. For comparisons between the two groups, a one-tailed *t*-test with a confidence level of 95% was used. If normality was not met, the non-parametric Mann–Whitney rank sum test was used.

## 3. Results

### 3.1. Human Fetal Spinal Cord NPCs Reside in the Ependymal Central Canal and the Spinal Parenchyma

Rodents maintain proliferative NSCs in the ependymal central canal, representing an endogenous source of cells for the repair of lesions [23,24,25]; however, we currently lack a detailed study of the human fetal spinal cord canal. Transmission electron microscopy (TEM) images (Figure 1A) demonstrated the organization of ependymal cells at a gestational stage corresponding to 20 weeks—a tight alignment of ependymal cells with a radial morphology organized as a pseudostratified epithelium forming the central canal. We observed features previously described in fetal mice spinal cord [26]—highly polarized and multiciliated ependymal cells with a large number of cilia in the apical zone facing the lumen of the central canal (white arrow), nuclei with condensed chromatin (*), and long large junction complexes (white arrowheads).

We explored the expression of sex determining region Y (SRY)-box 2 (SOX2), the earliest transcription factor expressed in neural stem/progenitor cells [27], which controls the specification of early neural lineages and brain and spinal cord development [28] as a means to identify spinal-cord-resident NPCs in fetal samples at the late gestational stage of 19–21.6 weeks. We found abundant positive nuclear staining for SOX2 in the spinal cord parenchyma and the cells of the central canal (Figure 1B). We also observed double-SOX2-/Ki67-positive cells in the central canal and distributed throughout the spinal cord parenchyma contributing to the active proliferative NPC population [29]. As the cell constituting the central canal migrate towards the dorsal and ventral regions of the spinal cord, they differentiate, specialize, and determine the spinal cord cytoarchitecture. In our samples, we identified elongating processes from both NESTIN- and GFAP-expressing cells from the roof plate of the central canal (Figure 1B), which may help to determine the dorso-ventral regionalization of the spinal cord, as previously reported in rodents [30,31]. We also observed neurons, oligodendrocytic precursors, and astrocytes positive for NEUN (red), OLIG2 (yellow), or GFAP (red), respectively, distributed through the spinal cord parenchyma, allowing the determination of dorso-ventral regionalization (Figure 1B).

### 3.2. hfNPCs Proliferate and Express Canonical Neural Markers in Vitro

NPCs self-renew, proliferate, and differentiate into the three neural lineages [32]; therefore, we evaluated the isolation and expansion of NPCs from whole human fetal spinal cord homogenates in neurosphere-like-forming cultures that support the self-renewal and clonal-like growth of NPCs [33]. Dissociation and culture of a human fetal spinal cord supported the efficient generation of neurospheres in free-floating conditions after two days of in vitro culture in the presence of the human recombinant mitotic factors bFGF, EGF, and LIF. After three days of culture, we observed an enrichment of primary (white arrow) and secondary (black arrow) neurospheres (Figure 2A).

After three days of culture, we expanded hfNPCs in adherent conditions in laminin-coated wells, retaining their morphology during the long-term expansion of subcultures (Figure 2B) as previously described for adult human NPCs [34]. Following subculture and reaching 80% confluency, we calculated the cell doubling time in hours (*t*) for fhNPCs for four passages (Figure 2C, black line). The *t* of rat NPCs derived from E15.5 spinal cords was also evaluated as a reference sample (Figure 2C, red line). Although we observed more significant heterogeneity in the results during the first two passages, cell doubling times during four passages did not significantly differ, ranging from 75.68 ± 10.87 to 103.1 ± 16.23 h, more than three-times the time needed for the rat fetal cells to duplicate in culture (Figure 2C). Nevertheless, hfNPCs at passage 4 continued to proliferate, with 21.4 ± 3.3% of cells incorporating BrdU at the S phase and 37.5 ± 4.7% displaying positive staining for the Ki67 mitotic marker (Figure 2E). 

Previous studies of the subventricular zone of adult brain tissue revealed a pinwheel structure, which corresponds to the adult neurogenic niche [35]; furthermore, this structure also represents a hallmark of stemness in mouse-spinal-cord-derived neurospheres [36]. To study if hfNPCs from late gestational stages also form neurogenic niche-like structures, we immunostained for γ-tubulin to indicate microtubule-organizing centers, centrosomes, basal bodies, and β-catenin to delineate cell borders for the identification of pinwheel structures. For the first time, we revealed that in vitro cultures of spinal cord hfNPC-derived neurospheres adopt a pinwheel structure and display the neural features of fetal developmental stages (Figure 2D).

To ascertain whether amplified hfNPCs expressed canonical neural markers at passages 2 to 3, we immunoassayed samples for neurogenic locus notch homolog protein 1 (NOTCH1), paired-box protein 6 (PAX6), NESTIN, SOX2, Forkhead Box J1 (FoxJ1), doublecortin (DCX), and Neurogenin1 (Figure 2F). hfNPCs expressed high levels of nuclear NOTCH1 and PAX6 (90.5 ± 13.8% and 91.7 ± 4.4% positive cells, respectively)—activated NOTCH translocates to the nucleus to support stemness [37], while Pax6 regulates NPC proliferation and self-renewal [30]. hfNPCs also stained positive for NESTIN (83.4 ± 2% positive cells) and SOX2 (79.7 ± 2.2% positive cells), which are expressed in late NPCs [38,39]. Additionally, 65 ± 6.4% of hfNPCs stained positive for FOXJ1, which is expressed in ciliated cells such as the ependymal cells [40], while hfNPCs also expressed DCX (56.4 ± 7.4%) and Neurogenin1 (45.8 ± 9.5%), with both proteins involved in neurogenesis [41,42] (Figure 2F).

### 3.3. PGA-SS-FAS Priming Enhances the Neuronal and Oligodendroglial Differentiation of hfNPCs

We next assessed the multipotent differentiation potential of hfNPCs after mitogen withdrawal and serum supplementation in the absence and presence of PGA-SS-FAS priming via immunofluorescence analysis for *β*-III-tubulin (pan-neuronal marker; green), GFAP (astrocyte marker; red), OLIG2 (oligodendrocyte marker, green), and NEUN (mature neuronal marker, red) (Figure 3A). Immunofluorescence quantitative analysis demonstrated that hfNPCs gave rise to neurons (9.85 ± 1.91 of cells positive for *β*-III-tubulin), astrocytes (17.78 ± 3.49% of cells positive for GFAP), and oligodendrocytes (15.54 ± 2.54% of cells positive for OLIG2) (Figure 3B), revealing the multipotency of the hfNPCs after in vitro amplification. Interestingly, PGA-SS-FAS priming significantly increased neuronal differentiation (up to four-times the number of *β*-III-tubulin-positive neurons in comparison with non-primed control hfNPCs; 43.61 ± 4.92% vs. 9.85 ± 1.91) and promoted mature neuronal differentiation (4.75 ± 1.11% NeuN-positive cells compared to non-primed hfNPCs, which lacked mature neurons) (Figure 3B). PGA-SS-FAS priming also significantly increased oligodendrocyte differentiation (30.11 ± 3.78% OLIG2-positive cells compared to non-treated hfNPCs), although we observed no differences regarding astrocyte differentiation (Figure 3B).

SCI activates the Rho/ROCK pathway, constituting one of axonal regrowth’s most significant inhibitory signals [43]. We activated the Rho/ROCK pathway by culturing hfNPCs in the presence of 10 μM LPA, a potent mitogen that induces neurite collapse [44], for 24 h, mimicking the intrinsic mechanism that blocks axon regeneration following SCI (Figure 3C,D). LPA induced a significant retraction of neurite-like processes in hfNPCs compared to control non-primed hfNPCs (28.45 ± 4.41% vs. 72.07 ± 8.96; Figure 3C, red arrows); however, PGA-SS-FAS priming of hfNPCs efficiently inhibited neurite-like retraction after LPA exposure (120.8 ± 8.16 vs. 28.45 ± 4.41%; Figure 3D, yellow arrows). These data suggest that priming with PGA-SS-FAS would prevent neuronal-like retraction of hfNPCs at the injury site following Rho/ROCK signaling activation. 

### 3.4. PGA-SS-FAS Priming Enhances the Ventral Engraftment of hfNPCs, Endogenous Neuronal Activation, and Neuronal Survival After Transplantation into the Injured Spinal Cord

We next performed an intramedullary transplant of hfNPCs primed with PGA-SS-FAS (hfNPCs + PGA-SS-FAS) or vehicle (hfNPCs) immediately after compressive injury to the eighth thoracic vertebrae segments in nude mice to evaluate their therapeutic potential (Figure 4). We did not observe significant differences in the total number of surviving hfNPCs when comparing hfNPCs and hfNPCs + PGA-SS-FAS (Figure 4A–C), in both cases with a low survival rate (with an estimated percentage for the total cells of 2.7 ± 1.1 for hfNPCs + PGA-SS-FAS and 2.6 ± 0.5 for the hfNPC group), as shown in the representative images of GFP-positive grafted cells (Figure 4A). hfNPCs primed with PGA-SS-FAS (hfNPCs + PGA-SS-FAS) possessed enhanced grafting and migratory capacities and were encountered from the ventral to the dorsal areas of the spinal cord, while non-primed cells were restricted to the dorsal and injected areas (Figure 4D). In addition, hfNPCs + PGA-SS-FAS grafts possessed increased expression of c-Fos (Figure 4E,F) (a marker associated with neuronal activity [45]) when compared to non-primed hfNPCs, thereby suggesting that PGA-SS-FAS priming enhances the activation of transplanted hfNPCs. To determine the safety of the hfNPC transplantation and the priming process, we evaluated the potential invasiveness of grafted cells for both groups outside the spinal cord. Four weeks after transplantation, we evaluated the expression of eGFP by transplanted hfNPCs in the brain, heart, and liver by semiquantitative PCR. We failed to detect GFP expression in any of the analyzed tissues (data not shown). GABAergic inhibitory Lbx1 and glutamatergic excitatory Tlx3 are transcription factors involved in neuronal fate determination of somatosensory interneuron populations located in the dorsal horns of the spinal cord, which modulate and integrate peripheral somatosensory inputs [46]. A low percentage of hfNPCs displayed Lbx1 and Tlx3 expression in the grafts, while priming with PGA-SS-FAS did not influence this cell fate determination, as we failed to find any significant differences between the groups (Figure 4G–I). 

Quantifying neuron survival at the injury site demonstrated no significant differences between primed and non-primed hfNPCs rostral or caudal to the injury and at the lesion epicenter (Figure 4J–L). Nevertheless, PGA-SS-FAS-primed hfNPCs showed a modest, but significant number of c-Fos^+^/NeuN^+^ cells at the injury site (Figure 4M), indicating a potential effect of the primed grafts on the subrounded neuronal activation. Furthermore, a modest increase in Lbx1 and Tlx3 interneurons surrounding the graft was found in the hfNPCs + PGA-SS-FAS group at the dorsal horn compared with the hfNPC group (Figure 4N–P). 

## 4. Discussion

The inflammatory and secondary damage milieu of SCI, which causes poor cell survival and grafting and improper differentiation, has been attributed as the factor limiting the therapeutic potential of cell therapy [47]. Survival rates and sufficient engraftment and integration into spinal cord circuits represent critical factors for successful cell transplantation and functional improvements [2]. In addition, aberrant neuronal connections induced by SCI can prompt allodynia, among other undesired effects [48]. Despite their multipotency, NPC transplants lack consistent neuronal motor differentiation [9,49], with differentiation generally driven towards the generation of glia by the SCI milieu. Combination therapies with varying levels of success have been developed to overcome these limitations—as reviewed by Griffin and Bradke [50]—with strategies including the combination of Notch inhibitors, to induce neuronal differentiation [51], with biocompatible matrices [52] or exogenous neurotrophic factors [53,54]. 

The origin and properties of NPCs have enormous significance in regenerating damaged spinal circuits [53]. Kadoya et al. recently demonstrated that rat fetal NPCs with spinal cord features (and not brain-derived NPCs) promoted regeneration of the corticospinal tract and functional motor improvement [53]. Moreover, Dulin et al. revealed that NPCs retain their features and differentiate towards specific phenotypes after transplantation into homologous regions of the host tissue, depending on their dorsal or ventral origin. [55]. 

Currently, translational strategies under investigation employ ESC- [16] or iPSC- [17] derived NSCs or immortalized fetal NPC lines [13] that have undergone significant manipulation prior to transplantation (e.g., reprogramming, long differentiation processes, or cell cycle manipulation for cell immortalization). We wanted to propose an alternative source for NPCs with minimal manipulation in the present study. To adhere to clinical-grade conditions, we isolated and in vitro expanded hfNPCs from 19.0- 21.6 weeks of gestation human fetal spinal cords under xeno-free conditions. We found that the isolated hfNPCs retained canonical neural features and multipotency in vitro, making them suitable as a clinical translational approach for SCI. In addition, we evaluated a combinatorial strategy by priming hfNPCs with an in-house-developed polymer-conjugate of the Rho/ROCK inhibitor fasudil (PGA-SS-FAS), which previously demonstrated neurogenerative properties in an immunocompetent rat model of SCI [19]. This approach allowed for the standardized culture of spinal-cord-derived hfNPCs and the generation of a procedure for improved combinatorial therapy suitable for cell replacement therapy in the injured spinal cord. 

We encountered SOX2-expressing proliferative NPCs densely packed within the central canal, but also occupying most of the spinal cord, giving rise to a heterogeneous population of progenitors at the primary culture expanded from the whole fetal spinal cord. Nevertheless, the heterogeneous hfNPC population possesses a homogeneous-like cell morphology in culture with a consistent cell fate profile when comparing different samples. hfNPCs retained stemness, with more than 80% of cells expressing SOX2 over several passages in vitro; however, we uncovered an important barrier for cell banking: hfNPCs possessed a low amplification efficiency, with a PD time more than three-times the time needed for cell duplication when compared with other cell populations derived from earlier gestational stages [12]. Although fhNPCs retained their proliferative capacity under the evaluated conditions, the employed growth-factor-enriched formulation (supplemented with bFGF, EGF, and LIF) requires further improvement to promote higher in vitro proliferative rates and cell expansion to make cell banking more feasible since rat-derived NPCs, also from late gestational stages, displayed a significant lower doubling time in the same cell culture conditions. Direct comparative analyses between different gestational ages for NPC isolation, expansion, and banking production will be required. 

In vitro priming with PGA-SS-FAS promoted faster cell maturation, favoring neuronal and oligodendroglial differentiation without interference with astroglial differentiation within 24 h of incubation. Poor neuronal differentiation has been reported as a significant limitation of NPC transplantation approaches; we now report a one-step priming procedure to partially overcome this problem, avoid in vivo application of PGA-SS-FAS, and reduce the potential side effects on host tissue. The convergence RhoA/ROCK pathway becomes activated by inhibitory molecules within the SCI milieu and plays a central role in inflammation, apoptosis, neuronal degeneration, and axon retraction [56]. Given the importance of these processes to SCI pathogenesis and the impairment of functional recovery, several pharmacological strategies have been developed to prevent cell death and promote axonal regeneration and functional recovery after SCI [57,58]. We previously described that NPCs derived from fetal rat spinal cords displayed improved neurite regrowth in vitro following treatment with PGA-SS-FAS, which also enhanced engraftment, induced a neuronal-like morphology, and elongated neurons from NPCs in vitro and in vivo [19]. Conversely, Stern et al. reported that RhoA inhibition had opposing roles in neurons and astrocytes, with RhoA activation limiting astrogliosis and RhoA ablation enhancing axon regeneration in neurons [59]. Considering these results, selective cell-specific approaches will be required to avoid side effects in host tissues.

Following SCI, axonal disruption and neuron death cause irreversible functional losses; therefore, preserving neuronal circuits, replacing lost neurons, and providing a regenerating environment to enhance plasticity after injury represent critical objectives for SCI therapeutics. We found that primed hfNPCs significantly preserved Tlx3- and Lbx1-expressing neuronal cells, which could be supported by the higher percentage of oligodendrocyte precursors induced by the treatment with PGA-SS-FAS prior to transplantation demonstrated in vitro. The Tlx3 homeobox gene functions as a developmental regulator of excitatory neurons, promoting glutamatergic excitatory specification and suppressing GABAergic specification in dorsal spinal cord neurons [60]. Lbx1 is required for the correct specification of early dorsal interneuron populations and plays a critical role in developing the spinal cord sensory pathways that transmit pain and touch [61]. Furthermore, grafted hfNPCs without the influence of the PGA-SS-FAS priming differentiated into Tlx3 and Lbx1 interneurons, showing phenotypically appropriate host target regions in the dorsal area of the spinal cord. 

PGA-SS-FAS-primed hfNPCs induced the in vivo expression of c-Fos, a classical marker of neuronal activity [62], which is also related to synaptic plasticity and learning [63]. We observed a double effect: First, PGA-SS-FAS priming activated grafted hfNPCs, which could influence the differential migratory profile encountered compared with the non-primed hfNPCs. Fasudil-induced migration has been previously described via activation of the MAPK signaling pathway in mesenchymal stem cells [64] and the ERK signaling pathway in microglia [65]. Further analysis of the mechanism of action involved in hfNPC migration needs further investigation. Second, the transplantation of PGA-SS-FAS-primed hfNPCs induced the increased activation of endogenous neurons surrounding the graft (measured by the increased expression of c-Fos in host neurons), which has been related to synaptic plasticity and learning in other systems [63]. Nonetheless, despite the identified histological signs showing modest, but significant, improved neural preservation capabilities on the hfNPCs + PGA-SS-FAS group, we did not find differences on the locomotion recovery by the BMS test weekly analysis (data not shown). 

## 5. Conclusions

Overall, our results provide evidence that PGA-SS-FAS-primed hfNPCs exert modest, but significant improved neuroprotective and a more migratory engraftment capability with an increased activation of surrounding endogenous neurons, which could provide a new combinatorial approach in a single formulation, which may serve as an improved cell therapy for SCI. However, since the differences reported here, employing PGA-SS-FAS-primed hfNPCs, did not improve graft survival and did not improve functional regeneration, we expect that PGA-SS-FAS-primed hfNPCs from earlier fetal stages would provide better results. In addition, since an immune-deprived mouse model was employed hosting the exogenously transplanted human cells, further evaluation in an immune competent model will be needed to address the important limitation of the immune rejection prior to clinical application. 

## Figures and Tables

**Figure 1 cells-11-03304-f001:**
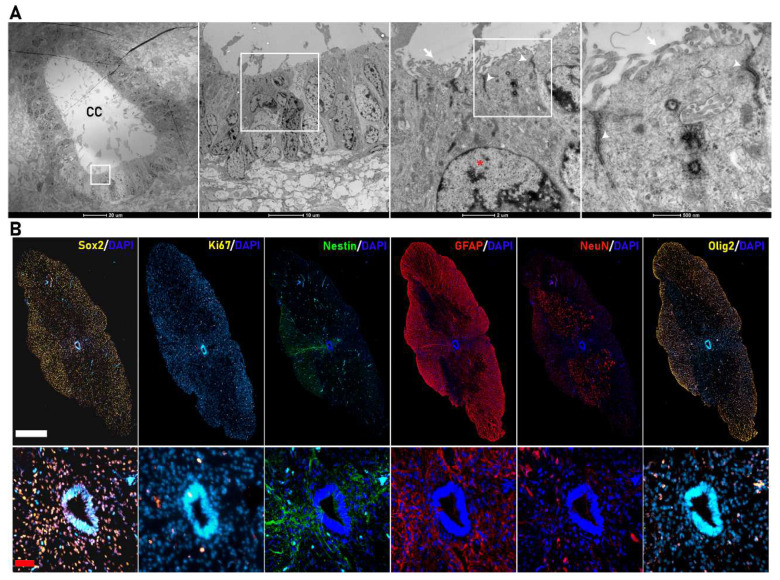
Cytoarchitecture of the human fetal spinal cord. (**A**) Transmission electron microscopy (TEM) images of the central canal of the human fetal spinal cord (scale bar (from left to right) = 20 µm, 10 µm, 2 µm, and 500 nm). The central canal is formed by ependymal cells organized as a pseudostratified epithelium with a large number of cilia in the apical zone (white arrow), nuclei with condensed chromatin (*), and large apical junction complexes (white arrowheads). (**B**) Representative coronal immunostaining images (complete coronal sections in upper panels and magnified view of the corresponding CC area in the lower panels) of human fetal spinal cords for SOX2 (yellow), Ki67 (yellow), Nestin (green), Gfap (red)**,** NeuN (red), and Olig2 (yellow) (white bar scale= 500 µm; red bar scale = 50 µm).

**Figure 2 cells-11-03304-f002:**
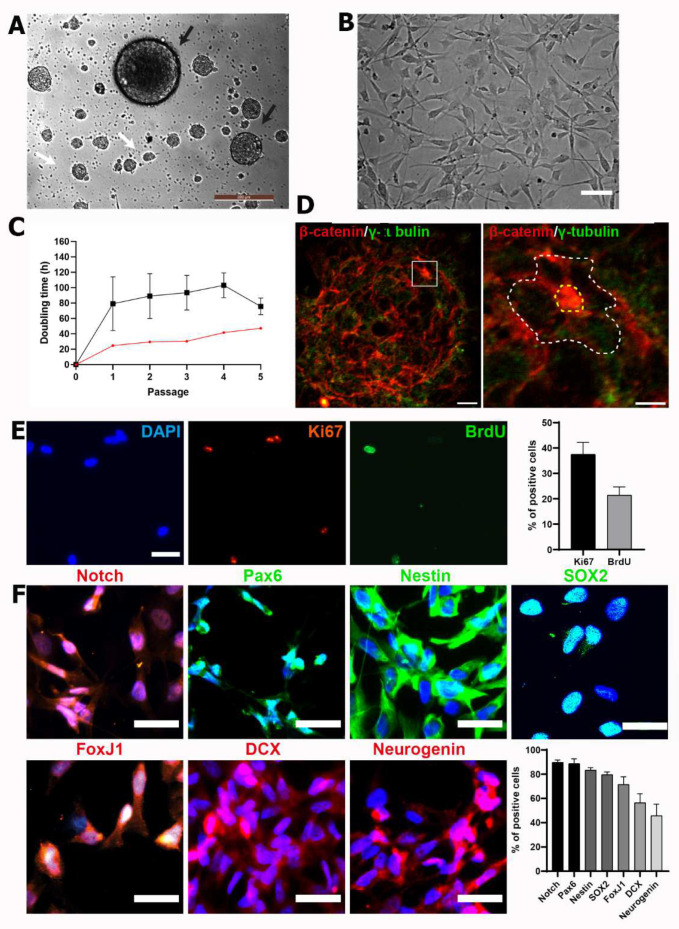
hfNPCs display proliferative potential and express canonical NPC markers in vitro. (**A**) Phase contrast image from hfNPCs as neurospheres-like cultures forming primary (black arrows) and secondary (white arrows) neurospheres as a hallmark of self-renewal in the presence of mitogens at passage 1. Scale bar = 200 µm. (**B**) Representative phase contrast image of hfNPC culture in adherent conditions in laminin-coated wells when cultured at passages 2–5 (Scale bar = 20 µm). (**C**) Doubling time (*t*) analysis of hfNPCs (black line; data presented as the mean ± SEM (*n* = 5 samples)) and one rat fetal NPC sample (red line) growing in adherent conditions over five passages. (**D**) The pinwheel cytoarchitecture (indicated with a white dotted line) of hfNPC at passage 1 from neurosphere-like cultures is highlighted after β-catenin (red; delimiting the cell perimeter) and γ-tubulin (green; for cilia detection) staining (scale bar = 20 µm (left panel) and 10 µm (right panel)). (**E**) Representative images from nuclear staining of DAPI (blue), Ki67 (red), and BrdU (green) of hfNPCs in adherent conditions (scale bar = 10 µm). Right panel: Quantification of the percentage of positive cells for BrdU and Ki67. Data presented as the mean ± SEM (*n* = 3 samples). (**F**) Left panel: Representative images of immunofluorescent staining for Notch, Pax6, Nestin, Sox2, FoxJ1, DCX, and neurogenin (scale bar = 20 µm). Right panel: Quantification of the percentage of positive cells for each canonical neural progenitor cell marker. Data presented as the mean ± SEM (*n* = 5 samples).

**Figure 3 cells-11-03304-f003:**
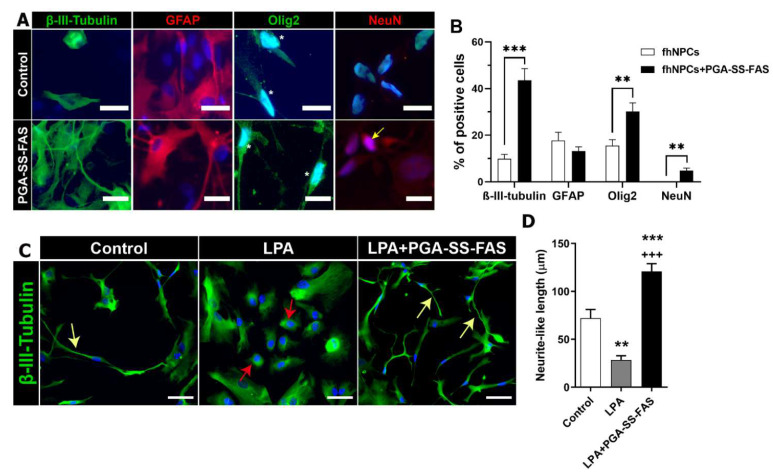
PGA-SS-FAS priming process enhances neuronal and oligodendrocyte differentiation and induces neurite outgrowth in a model of neurite retraction. (**A**) Representative immunofluorescence images of hfNPCs using *β*-III-tubulin for immature neurons (green), GFAP for astrocytes (red), Olig2 for oligodendrocytes (green), NeuN for mature neurons (red), and DAPI (blue) (scale bar = 20 µm) (positive nuclear staining is indicated with an arrow) in the presence of the PGA-SS-FAS compound for inducing the priming process (PGA-SS-FAS) or its vehicle (control). (**B**) Quantification of the percentage of positive cells for each of the indicated cell markers with (black bars) or without (white bars) 24 h PGA-SS-FAS priming. Data presented as the mean ± SEM determined by Student’s unpaired t-test (*n* = 3). ** *p* < 0.01, *** *p* < 0.001 vs. hfNPCs. (**C**) Representative immunofluorescence images of *β*-III-tubulin (green) and DAPI (blue) staining during the in vitro neurite retraction assay induced by lysophosphatidic acid (LPA) treatment during 24 h in hfNPC adherent cultures. Yellow arrows indicate neurite-like processes emanating from neuronal progenitors in the control condition (in the absence of LPA) and in co-treated cultures (LPA + PGA-SS-FAS); red arrows indicate rounded cells 24 h after LPA treatment and induced neurite-like retraction (scale bar = 50 µm). (**D**) Neurite length quantification presented as the mean ± SEM determined by one-way ANOVA with the Tukey multiple comparison test (*n* = 3). ** *p* < 0.01, *** *p* < 0.001 vs. control; +++ *p* < 0.001 vs. control.

**Figure 4 cells-11-03304-f004:**
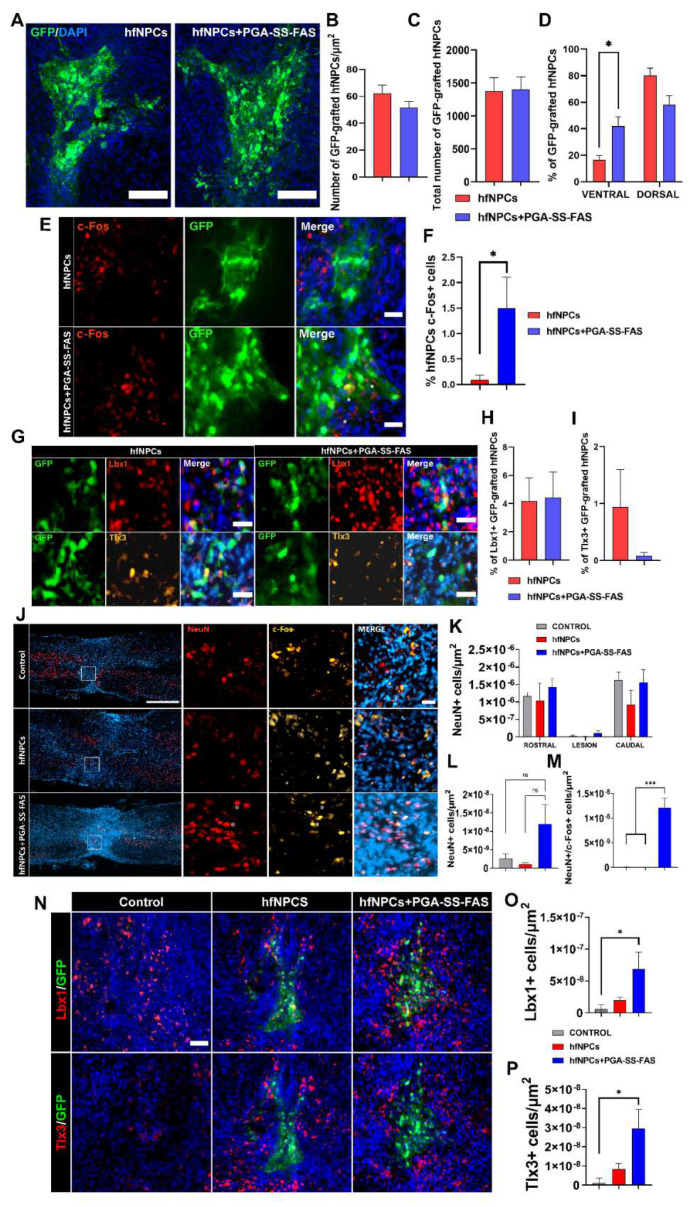
Quantification, distribution, and phenotypic characterization of non-primed and primed hfNPCs post-transplantation. (**A**) Representative images of GFP staining for hfNPC-grafted cells in spinal cord horizontal sections for the indicated groups (green) (scale bar = 50 µm). (**B**) Quantification of GFP-positive transplanted cells normalized to the total measured thickness in µm^2^. (**C**) quantification of GFP-positive transplanted cells in absolute numbers every 5th 20 mm section. (**D**) Quantification of the dorso-ventral distribution of the GFP-positive cells expressed in percentage of the total number of quantified cells. (**E**) representative images of c-Fos+ (red) and GFP (green) and the merged image including DAPI (blue) -positive immunostaining of grafted GFP-positive hfNPCs. (**F**) Quantification of the percentage of c-Fos-positive hfNPCs in the primed or non-primed groups, presented as the mean ± SEM and determined by Student’s unpaired t-test (*n* = 3). (* *p* < 0.05). (**G**) Representative images of immunofluorescent staining for GFP- (green), Lbx1- (red), and Tlx3- (yellow) positive grafted cells and merged images with DAPI (blue) of both experimental groups (scale bar = 25 µm). (**H**) Quantification of the percentage of Lbx1. (**I**) Tlx3-positive hfNPCs presented as the mean ± SEM and determined by Student’s unpaired t-test (*n* = 3)). (**J**) Representative immunostainings of c-Fos (yellow), NeuN (red), and DAPI (blue) (white scale bar = 500 µm; red scale bar = 25 µm). (**K**) Quantification of NeuN-positive cells in the rostral, lesion normalized to the total measured thickness in µm^2^, and caudal sites. (**L**) Quantification of NeuN-positive cells in the lesion site; data presented as the mean ± SEM determined by one-way ANOVA with Tukey’s multiple comparison test (*n* = 3). (**M**) Quantification of NeuN/c-Fos-double-positive cells in the lesion site normalized to the total measured thickness in µm^2^. Data presented as the mean ± SEM determined by one-way ANOVA with Tukey’s multiple comparison test (*n* = 3). (**N**) Representative immunostainings of (top panel) Lbx1 (red), GFP (green), and DAPI (blue) and (bottom panel) Tlx3 (red), GFP (green), and DAPI (blue) (scale bar = 50 µm) for the indicated experimental groups. (**O**) Quantification of Lbx1. (**P**) Tlx3-positive cells in the area surrounding the graft and lesion normalized to the total measured thickness in µm^2^. Data presented as the mean ± SEM and determined by one-way ANOVA with Tukey’s multiple comparison test (*n* = 3) (* *p* < 0.05, *** *p* < 0.001 (* *p* < 0.05, *** *p* < 0.001, ns = non-significant).

## Data Availability

Not applicable.

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
