# Peer review of "Transplantation of Human-Fetal-Spinal-Cord-Derived NPCs Primed with a Polyglutamate-Conjugated Rho/Rock Inhibitor in Acute Spinal Cord Injury"

_cells, 2022, doi:10.3390/cells11203304_

Round 1

Reviewer 1 Report

Minor revisions

Methods:

Please, check throughout the whole text for typos (e.g., line 112 ultra-low attached --> ultra-low attachment)

Line 191 authors indicated the NeuronJ plug-in, but the indication of the software is missing. Is it ImajeJ? Which version?

Figures:

In general, figure legends quality should be improved, better describing the figures. For example:

Figure 2 - A: please indicate in the legend the why you added arrows and what they are indicating; D, second image: please indicate also the meaning of the dotted line

Figure 3 - A: please, indicate what is the meaning of the arrow 

Author Response

Reviewer 1:

Comments and Suggestions for Authors

Minor revisions

Methods:

Please, check throughout the whole text for typos (e.g., line 112 ultra-low attached --> ultra-low attachment):

We have revised the whole manuscript in pairs detecting several typos as the one highlighted by the reviewer; all detected typos have been properly corrected.  

Line 191 authors indicated the NeuronJ plug-in, but the indication of the software is missing. Is it ImajeJ? Which version?

We apologise for the missing indication. As the reviewer suggested, a complete indication has been now included in Methods for the employed software, as followed: “ NeuronJ plug-in from ImageJ v1.48”

Figures:

In general, figure legends quality should be improved, better describing the figures. For example:

Figure 2 - A: please indicate in the legend the why you added arrows and what they are indicating; D, second image: please indicate also the meaning of the dotted line

Figure 3 - A: please, indicate what is the meaning of the arrow 

We have improved the legend description for all four figures, including all reviewer´s suggestions.  

Reviewer 2 Report

This paper describes the technologically challenging task of preparing and characterizing human fetal neural precursors for spinal cord lesion therapy by intramedullary transplantation. The authors convincingly demonstrate the isolation and culture of these NPCs from aborted human fetuses and present data supporting their potential for neuronal differentiation, confirming their bibliographically documented expertise in this area. In these human NPCs they also reproduce the beneficial effects of a chemically enhanced form of the RhoA/ROCK inhibitor fasudil on survival and neuronal differentiation, which they previously documented for rat NPCs. Finally, data is presented from transplants in SCI lesions of nude mice, showing some promising preliminary data indicating possible therapeutic potential for SCI.

In general, the work has been carried out methodically and carefully, and the authors have made a significant advance in this difficult area of translational biomedicine with their in vitro-primed human NPCs. The main criticism I have is for the in vivo data presented in Figure 4 – while the results are promising given the very low numbers of positive cells counted, and the extremely small percentage of the total grafted cells that these represent (1% or less for Fos and Thx3), the authors should be more cautious in their conclusions: “PGA-SS-FAS primed hfNPCs displayed significantly enhanced neuronal activity, as measured by the number of c-Fos+/NeuN+ cells in the injury site”; “we found that the hfNPCs+PGA-SS-FAS group displayed an increase in Lbx1 and Tlx3 interneurons surrounding the graft at the dorsal horn”; “PGA-SS-FAS-primed hfNPCs induced the increased activation of endogenous neurons surrounding the graft (measured by the increased expression of c-Fos in host neurons)”. There is no description of how many total positive cells were counted in how many samples – although it is understood that such in vivo work can only be semi-quantitative, this would help give the reader an idea of the effect size. It would also be helpful to have some kind of estimate of what percentage of the hfNPCs injected actually survive in the grafts after one month. Given the difficulty of these xeno-transplant SCI experiments, the authors should also mention possible interference of the animal model to explain the differences compared to their previous results with rat NPCs in the rat SCI model.

Legends and figures should be improved, especially Fig 4. Some examples: DAPI (blue) is missing in the legends 3A, 4A, 4D, 4F and 4I; Panel 4A y axis label missing the squared sign of μm2; Legend Fig 4F – “grated” should be “grafted”; Legend Fig 4H – remove extra “ ); ” at the end; Legend Fig 4I – there is no red scale bar; Panels 4J, 4K, 4N and 4O – the negative exponents are too small to read, and cell counts should be expressed as whole cells per mm2 or cm2 rather than tiny fractions of cells per μm2. (See above note on methodology and conclusions for Fig4.)

Some examples of minor language errors: L425 replace “…we next intramedullary transplanted…” with “…we next performed and intramedullary transplant of…”; L432 replace “…spinal cord meanwhile, we encountered non-primed cells restricted to…” with “…spinal cord while non-primed cells were restricted to…”; L438 replace “outwith” with “outside”.

Author Response

Reviewer 2:

Comments and Suggestions for Authors

This paper describes the technologically challenging task of preparing and characterizing human fetal neural precursors for spinal cord lesion therapy by intramedullary transplantation. The authors convincingly demonstrate the isolation and culture of these NPCs from aborted human fetuses and present data supporting their potential for neuronal differentiation, confirming their bibliographically documented expertise in this area. In these human NPCs they also reproduce the beneficial effects of a chemically enhanced form of the RhoA/ROCK inhibitor fasudil on survival and neuronal differentiation, which they previously documented for rat NPCs. Finally, data is presented from transplants in SCI lesions of nude mice, showing some promising preliminary data indicating possible therapeutic potential for SCI.

In general, the work has been carried out methodically and carefully, and the authors have made a significant advance in this difficult area of translational biomedicine with their in vitro-primed human NPCs.

The main criticism I have is for the in vivo data presented in Figure 4 – while the results are promising given the very low numbers of positive cells counted, and the extremely small percentage of the total grafted cells that these represent (1% or less for Fos and Thx3), the authors should be more cautious in their conclusions: “PGA-SS-FAS primed hfNPCs displayed significantly enhanced neuronal activity, as measured by the number of c-Fos+/NeuN+ cells in the injury site”; “we found that the hfNPCs+PGA-SS-FAS group displayed an increase in Lbx1 and Tlx3 interneurons surrounding the graft at the dorsal horn”; “PGA-SS-FAS-primed hfNPCs induced the increased activation of endogenous neurons surrounding the graft (measured by the increased expression of c-Fos in host neurons)”.

There is no description of how many total positive cells were counted in how many samples – although it is understood that such in vivo work can only be semi-quantitative, this would help give the reader an idea of the effect size. It would also be helpful to have some kind of estimate of what percentage of the hfNPCs injected actually survive in the grafts after one month.

Following reviewer´s comments and suggestions we have modified our conclusions from results shown and described in Figure 4 based on the low quantified and identified preserved neurons. A total number of quantified surviving grafted cells has been now included in Figure 4C. The total numbers of counted cells was performed from every 5th section to the total analysed tissue thickness in mm2 for each of the 3 analysed animals in every experimental group; now indicated in material and methods.  

Given the difficulty of these xeno-transplant SCI experiments, the authors should also mention possible interference of the animal model to explain the differences compared to their previous results with rat NPCs in the rat SCI model.

We have included additional statements regarding the potential interference of the animal model in the discussion section following reviewer suggestions.

Legends and figures should be improved, especially Fig 4. Some examples: DAPI (blue) is missing in the legends 3A, 4A, 4D, 4F and 4I; Panel 4A y axis label missing the squared sign of μm2; Legend Fig 4F – “grated” should be “grafted”; Legend Fig 4H – remove extra “ ); ” at the end; Legend Fig 4I – there is no red scale bar; Panels 4J, 4K, 4N and 4O – the negative exponents are too small to read, and cell counts should be expressed as whole cells per mm2 or cm2 rather than tiny fractions of cells per μm2. (See above note on methodology and conclusions for Fig4.)

We have corrected all identified mistakes in all Figures and their corresponding legends. The size of the font of the negative exponents that were difficult to read have been increased. However, we have not modified the cell counts as suggested, since the relative changes will not be modified and expressed in μm2 will indeed reduce the exponential numbers.

Some examples of minor language errors: L425 replace “…we next intramedullary transplanted…” with “…we next performed and intramedullary transplant of…”; L432 replace “…spinal cord meanwhile, we encountered non-primed cells restricted to…” with “…spinal cord while non-primed cells were restricted to…”; L438 replace “outwith” with “outside”.

We have revised and corrected all typos and language errors through the whole manuscript included the ones highlighted by the reviewer.

Reviewer 3 Report

The manuscript titled “Transplantation of Human Fetal Spinal cord-derived NPCs Primed with a Polyglutamate-conjugated Rho/Rock Inhibitor as a Therapy for Acute Spinal Cord Injury” confirmed that human fetal spinal cord NPCs (hfNPCs) resided in the Ependymal Central Canal and the Spinal Parenchyma. hfNPCs from legal gestational interruptions at 19-21 weeks maintain the characteristics of proliferation and multipotential differentiation (e.g., neurons, astrocytes, oligodendrocytes) in vitro, which means that it is feasible to be a cell bank for SCI treatment. PGA-SS-FAS priming hfNPCs exhibit efficient differentiation into neurons and oligodendrocytes. Moreover, the transplantation of primed hfNPCs into the acute SCI mouse model prompted increased preservation of GABAergic inhibitory Lbx1 expressing and glutamatergic excitatory Tlx3-expressing somatosensory interneurons, and elevated numbers of preserved, c-Fos expressing, activated neurons surrounding the injury epicenter.

Overall, this is an interesting study and the authors have performed multiple experiments. However, the study has multiple inadequacies that severely limited its quality.

1. The advantage of hfNPC as a cell bank for SCI treatment was not demonstrated. As an innovative cell used in this research, the proliferation efficiency, differentiation difference, and safety should be compared with NPC derived from different periods in the fetal spinal cord and other sources of NPC.

2. The histological data after hfNPC transplantation were obtained on the animal model of nude mice, avoiding the problems caused by immune rejection. However, the immune response of exogenous cell implantation is an inevitable factor that limits the application of human fetal-derived NPC as a therapeutic cell resource.

3. Obvious regeneration of newborn neurons and tissue repair were observed in the PGA-SS-FAS-primed hfNPCs transplantation group at the histological level. However, whether primed hfNPCs could significantly improve motor or sensory recovery after SCI remains unknown. Therefore, animal behavioral tests are crucial to support the conclusions of this study.

4. A better and clearer picture of SOX2 staining should be provided in Fig.2F.

5. The high background of immunofluorescence images needs to be adjusted to improve image quality in Fig.3A.

Author Response

Reviewer 3:

Comments and Suggestions for Authors

The manuscript titled “Transplantation of Human Fetal Spinal cord-derived NPCs Primed with a Polyglutamate-conjugated Rho/Rock Inhibitor as a Therapy for Acute Spinal Cord Injury” confirmed that human fetal spinal cord NPCs (hfNPCs) resided in the Ependymal Central Canal and the Spinal Parenchyma. hfNPCs from legal gestational interruptions at 19-21 weeks maintain the characteristics of proliferation and multipotential differentiation (e.g., neurons, astrocytes, oligodendrocytes) in vitro, which means that it is feasible to be a cell bank for SCI treatment. PGA-SS-FAS priming hfNPCs exhibit efficient differentiation into neurons and oligodendrocytes. Moreover, the transplantation of primed hfNPCs into the acute SCI mouse model prompted increased preservation of GABAergic inhibitory Lbx1 expressing and glutamatergic excitatory Tlx3-expressing somatosensory interneurons, and elevated numbers of preserved, c-Fos expressing, activated neurons surrounding the injury epicenter.

Overall, this is an interesting study and the authors have performed multiple experiments. However, the study has multiple inadequacies that severely limited its quality.

We appreciate reviewer´s comments, and in accordance we have followed all suggestions as indicated below:

1.The advantage of hfNPC as a cell bank for SCI treatment was not demonstrated. As an innovative cell used in this research, the proliferation efficiency, differentiation difference, and safety should be compared with NPC derived from different periods in the fetal spinal cord and other sources of NPC.

In order to compare the proliferative rates found within the human samples, we have included in Figure 2C the population doubling data obtained from rat fetal samples, in both cases derived from late fetal developmental stages.

  1. The histological data after hfNPC transplantation were obtained on the animal model of nude mice, avoiding the problems caused by immune rejection.However, the immune response of exogenous cell implantation is an inevitable factor that limits the application of human fetal-derived NPC as a therapeutic cell resource.

We have included this observation/comment of the reviewer as part of the conclusions, evidencing the need of further evaluation of the immune rejection limitation and the therapy translation.    

  1. Obvious regeneration of newborn neurons and tissue repair were observed in the PGA-SS-FAS-primed hfNPCs transplantation group at the histological level. However, whether primed hfNPCs could significantly improve motor or sensory recovery after SCI remains unknown. Therefore, animal behavioral tests are crucial to support the conclusions of this study.

We had performed functional locomotion analysis using the BMS scale in an open field, however no significant differences were detected. We have included this data in result section and added additional discussion regarding this limitation.   

  1. A better and clearer picture of SOX2 staining should be provided in Fig.2F.

We have substituted the SOX2 image for a better representation in Fig.2F

  1. The high background of immunofluorescence images needs to be adjusted to improve image quality in Fig.3A.

As reviewer suggested, Fig.3A has been adjusted to reduce background and improve image quality

Round 2

Reviewer 3 Report

This study clarified that human spinal cord-derived neural progenitor cells primed with PGA-SS-FAS increased the efficiency of differentiation into neurons and oligodendrocytes. After transplantation into nude mice with spinal cord injury, there was no significant change in cell survival, but the efficiency of migration to the dorsal and injured areas increased, and the activation of surrounding neurons was promoted. 

As described in lines 468-479, PGA-SS-FAS did not affect the proportion of hfNPC differentiated into Lbx1 and Tlx3 positive interneurons but recruited more neurons around the transplanted cells. The specific mechanism is not clearly explained in the discussion. If only to promote neuronal activation, cell transplantation strategy is not so necessary. 

In Fig.4G, immunostaining showed that the proportion of PGA-SS-FAS primed hfNPC differentiated into Lbx1 positive neurons was significantly higher, while the Tlx3 positive neurons did not seem to be significantly different, which was not consistent with the statistical results in Fig.4H and Fig.4I. 

Finally, the authors state that this approach did not improve significantly in animal behavioral experiments, which also suggests that although PGA-SS-FAS primed hfNPC transplantation has many advantages, it is inadequate as a therapy for acute spinal cord injury.

Author Response

Reviewer 3, round #2

Comments and Suggestions for Authors

This study clarified that human spinal cord-derived neural progenitor cells primed with PGA-SS-FAS increased the efficiency of differentiation into neurons and oligodendrocytes. After transplantation into nude mice with spinal cord injury, there was no significant change in cell survival, but the efficiency of migration to the dorsal and injured areas increased, and the activation of surrounding neurons was promoted. 

As described in lines 468-479, PGA-SS-FAS did not affect the proportion of hfNPC differentiated into Lbx1 and Tlx3 positive interneurons but recruited more neurons around the transplanted cells. The specific mechanism is not clearly explained in the discussion. If only to promote neuronal activation, cell transplantation strategy is not so necessary. 

We found that hfNPC transplanted cells could provide a source of interneurons positive for Lbx1 and Tlx3, as showed in Figure 4G-I. However, the PGA-SS-FAS priming procedure did not influence on its differentiation process. Nevertheless, the PGA-SS-FAS primed transplanted hfNPCs significantly preserved a higher number of host interneurons and increased the total number of activated neurons co-localizing with c-fos. This phenomena could be in part supported by the higher percentage of oligodendrocyte precursors provided by the primed hfNPCs grafts, as demonstrated in vitro.

In Fig.4G, immunostaining showed that the proportion of PGA-SS-FAS primed hfNPC differentiated into Lbx1 positive neurons was significantly higher, while the Tlx3 positive neurons did not seem to be significantly different, which was not consistent with the statistical results in Fig.4H and Fig.4I. 

Selective images in Fig4G were not representative, we apologize for this mistake. We have selected and incorporated a new image to better represent hfNPC group in Fig. 4G.

Finally, the authors state that this approach did not improve significantly in animal behavioral experiments, which also suggests that although PGA-SS-FAS primed hfNPC transplantation has many advantages, it is inadequate as a therapy for acute spinal cord injury

We have included an extra conclusion remarking the limitation of the reported strategy on functional regeneration, hypothesizing as a potential reason, on the low survival rates got in both transplanted groups.  It has been vastly described that NPC transplantation prompts neuroprotection and induces neurodegeneration in animal models (Assinck P et al. Nat Neurosci 2017, 20, 637-647, doi:10.1038/nn.4541.). However,  survival rates and sufficient engraftment and integration into spinal cord circuits represent critical factors for successful functional improvements. Results presented here aimed to explore a clinical approach for improving cell therapy in SCI employing low number of transplanted cells in a nude mice.  Since previous reports employing human neural stem cells with earlier developmental stages showed the most successful results, with high survival rates and functional recovery  (Lu P et al, Cell. 2012 Sep 14;150(6):1264-73; Rosenzweig ES et al, Nat Med. 2018 May;24(4):484-490),  we propose to get advantage of the PGA-SS-FAS primed procedure but employing NPCs from earlier developmental stages.
